# Teropavimab and zinlirvimab sensitivity in people living with multidrug-resistant HIV-1: data from the PRESTIGIO Registry

Vincenzo Spagnuolo,[1,2] Laura Galli,[1] Jiani Li,[3] Keith Dunn,[3] Filippo Lagi,[4] Roberta Gagliardini,[5] Loredana Sarmati,[6] Anna Maria Cattelan,[7] Andrea Giacomelli,[8] Maria Mercedes Santoro,[9] Maurizio Zazzi,[10] Christian Callebaut,[3] Antonella Castagna,[1,2] Laurie A. VanderVeen[3]

**ABSTRACT** We characterized sensitivity to teropavimab (TAB) and zinlirvimab (ZAB) in people living with four-class drug-resistant HIV (4DR-PWH). This was a multicenter, observational study using plasma or peripheral blood mononuclear cells collected from 50 4DR-PWH (25 with HIV-1 RNA > 1,000 copies/mL matched by age, sex, nadir CD4+, and years on ART to 25 virologically suppressed [HIV-1 RNA < 50 copies/mL]) enrolled in the PRESTIGIO Registry (NCT04098315) with a documented 4DR (NRTIs, NNRTIs, PIs, and INSTIs). Phenotypic sensitivity to bNAbs was determined using the PhenoSense monoclonal antibody assay (Monogram), with susceptibility defined as $IC_{90} \leq 2$ µg/mL. The HIV-1 envelope was genotyped by next-generation sequencing, and sequences were analyzed for the presence of multi-position HIV-1 envelope amino acid signatures associated with *in vitro* phenotypic susceptibility to TAB and ZAB. Of 46/50 (92%) participants with PhenoSense mAb assay results, 35 (76%) were phenotypically sensitive to TAB, 23 (50%) to ZAB, and 19 (41%) to both bNAbs; seven (15%) were phenotypically resistant to both bNAbs. The proportion of individuals with sensitivity to both bNAbs was similar in participants with viremia (41%) and those with virologic suppression (42%; $P = 0.99$). We observed marginal correlations between TAB 90% inhibitory concentration ($IC_{90}$) values and years since HIV diagnosis at the time of sample collection (*Spearman* r = 0.29, $P = 0.05$) as well as between ZAB $IC_{90}$ values and CD8+ cell count (*Spearman* r = −0.32, $P = 0.05$). A significant number of the 4DR-PWH analyzed were found to have virus susceptible to TAB and ZAB. These data provide proof-of-concept that selected multidrug-resistant PWH may be candidates for future trials investigating bNAbs-containing regimens to achieve or maintain virologic suppression.

**IMPORTANCE** Multidrug-resistant HIV presents significant challenges for treatment, often leaving individuals with a limited range of therapeutic alternatives. This study provides crucial insights into the efficacy of two promising broadly neutralizing antibodies, teropavimab (TAB) and zinlirvimab (ZAB), in individuals living with HIV who have developed resistance to multiple drug classes. The findings indicate that a significant proportion of the population remains susceptible to these novel treatments, irrespective of their viral suppression status. These results offer a promising basis for developing new therapeutic strategies to improve outcomes for individuals with multidrug-resistant HIV who have a history of extensive treatment, paving the way for future clinical trials aimed at achieving long-term viral suppression with novel drug regimens.

**KEYWORDS** broadly neutralizing antibodies, teropavimab, zinlirvimab, heavily treatment experienced, drug resistance, HIV

Address correspondence to Vincenzo Spagnuolo, spagnuolo.vincenzo@hsr.it.

A.C. received personal fees for advisory boards, speaker panels, and educational materials from Gilead Sciences, ViiV Healthcare, Janssen-Cilag, and Theratechnlogies. V.S. received grants from Gilead Sciences, personal fees for speaker panels from Gilead Sciences, ViiV Healthcare, and Merck Sharp & Dohme. M.Z. received funding from Gilead Sciences, ViiV Healthcare, Theratechnologies, and Merck Sharp and Dohme for his institution, and personal fees from Gilead Sciences, ViiV Healthcare, GSK, and Merck Sharp and Dohme. M.M.S. received personal fees for advisory boards, speaker panels, and educational materials from Gilead Sciences, ViiV Healthcare, and Merck Sharp and Dohme. A.M.C. received fundings from Gilead Science, ViiV Healthcare, and GSK for her institution, and personal fees from Gilead Sciences, ViiV Healthcare, MSD, Janssen-Cilag, and Angelini. F.L. received personal fees from Merck Sharp and Dohme, Gilead Sciences, ViiV Healthcare, and Jansenn-Cilag. R.G. received personal fees for speaker panels from Gilead Sciences, ViiV Healthcare, and Merck Sharp & Dohme. A.G. received grants from Gilead Sciences, personal fees for speaker panels from Gilead Sciences, ViiV Healthcare, Janssen-Cilag, and Merck Sharp & Dohme. J.L., K.D., C.C., and L.A.V. are employees and shareholders of Gilead Sciences, Inc. L.G. has no conflict of interest to disclose.

See the funding table on p. 10.

This study was presented in part at the 2024 Conference of Retroviruses and Opportunistic Infections on 3–6 March 2024 in Denver, Colorado (poster number 691).

Harboring multidrug-resistant (MDR) virus is a clear risk factor for clinical progression and death in people living with HIV (PWH) (1, 2). In these individuals, it is often very difficult to initiate a suppressive regimen (in the case of virologic failure) or simplify an ongoing regimen (in the case of suppressive therapy) due to the unavailability of fully active drugs, previous antiretroviral therapy (ART)-related toxicities, or lack of adherence (3).

Therefore, the availability of drugs with novel mechanisms of action and long-lasting efficacy would represent a substantial advancement in therapy for this population.

Lenacapavir (LEN), a novel HIV-1 capsid inhibitor, is currently approved for twice-yearly dosing in heavily treatment-experienced (HTE) PWH in combination with other antiretrovirals and being studied with various compounds for long-acting oral and subcutaneous (SC) injection every 3–6 months (4).

Teropavimab (formerly GS-5423 or 3BNC117-LS; TAB) and zinlirvimab (formerly GS-2872 or 10-1074-LS; ZAB) are broadly neutralizing antibodies (bNAbs) that target non-overlapping HIV-1 envelope spike sites (CD4 binding site and V3 loop, respectively) and designed to have long half-lives, potentially allowing twice-yearly dosing (5, 6). TAB and ZAB and their parental antibodies 3BNC117 and 10-1074 can neutralize a significant proportion of diverse global HIV-1 strains, although the breadth of neutralization varies by subtype (7–10).

In addition, selective immune pressure during untreated infection may drive the HIV-1 envelope evolution, resulting in viral populations that can evade recognition by bNAbs and limit their effectiveness. Thus, bNAbs susceptibility testing before initiating treatment with TAB and ZAB may improve clinical success. While there is currently no standard for determining whether viral strains are susceptible to bNAbs, the genotypic and phenotypic evaluations of viral susceptibility have been explored in clinical trials of 3BNC117, 10-1074, and their long-acting derivatives (11–15).

In a phase 1b proof-of-concept study (NCT04811040) in chronically treated, virologically suppressed PWH who discontinued oral ART and initiated a regimen of SC LEN plus weight-based intravenous infusions of TAB and ZAB, 18/20 (90%) participants maintained virologic suppression at week 26 (15). In an ongoing phase 2 study (NCT05729568), efficacy of the every-6-months regimen of LEN with fixed doses of TAB and ZAB was similar to that of daily oral ART through week 26 (16). In both studies, which evaluated PWH susceptible to both TAB and ZAB, as determined by *in vitro* phenotype, the proportion of PWH susceptible to both bNAbs was approximately 50% (15, 16).

Interestingly, the 26 week efficacy of the combination of LEN plus TAB and ZAB appears to be maintained, even in the presence of susceptibility to either TAB or ZAB alone (17).

Given that HIV-1 susceptibility TAB and ZAB may vary across populations, future studies should aim to evaluate diverse cohorts. Although TAB and ZAB may be useful for the treatment of PWH harboring MDR strains, data in this specific group of PWH are limited. In this study, we characterized the susceptibility to TAB and ZAB and HIV-1 envelope diversity in people living with four-class drug-resistant HIV (4DR-PWH).

## RESULTS

Fifty 4DR-PWH were evaluated (25 viremic and 25 non-viremic). Phenotypic assay failure was observed in four individuals (three in the viremic group and one in the non-viremic group), allowing the analysis of TAB and ZAB susceptibility in a total of 46 PWH. The characteristics of the individuals with analyzable samples (median age 55 [interquartile range: 48–58] years; 80% male) were indicative of a long history of HIV-1 infection (26 [23–32] years), extensive treatment history (23 [21–27] years of antiretroviral therapy [ART], 10 [5–20] previous ART lines), and a history of significant immunosuppression (nadir CD4+: 36 [8–83] cells/µL). HIV subtype B was present in 31 (91%) 4DR-PWH evaluated, while HIV-1 RNA viral load was 4.17 (3.54-4.89) $\log_{10}$ copies/mL in viremic individuals.

No significant differences are observed between viremic and non-viremic 4DR-PWH with the exception of the CD4+ cell count (193 [111–289] cells/mcL versus 618 [488–866]; $P < 0.01$), CD4+/CD8+ ratio (0.25 [0.13–0.36] vs. 0.81 [0.47–0.96]; $P < 0.01$), and

**TABLE 1**  Demographic, virological, and therapeutic characteristics of PWH at the time of sample collection according to the HIV-RNA viral load[a]

| Variable | Category | Overall (n = 46) | HIV-RNA ≥ 1,000 cp/mL (n = 22) | HIV-RNA < 50 cp/mL (n = 24) | P-value[b] |
|---|---|---|---|---|---|
| Age (years) | | 55 (48–58) | 54 (32–58) | 55 (49–59) | 0.46 |
| Sex assigned at birth | | | | | 1.00 |
| | Female | 9 (20%) | 4 (18%) | 5 (21%) | |
| | Male | 37 (80%) | 18 (82%) | 19 (79%) | |
| HIV-1 tropism | | | | | 0.72 |
| | CCR5 | 16 (37%) | 8 (40%) | 8 (35%) | |
| | CXCR4 | 27 (63%) | 12 (60%) | 15 (65%) | |
| HIV-1 subtype | | | | | 0.51 |
| | B | 31 (91%) | 18 (95%) | 13 (86%) | |
| | CRF02_AG | 1 (3%) | 0 (0%) | 1 (7%) | |
| | F | 2 (6%) | 1 (5%) | 1 (7%) | |
| Years since HIV infection | | 25.8 (22.9–31.7) | 25.5 (22.3–31.4) | 26.6 (23.0–31.8) | 0.72 |
| Years of ART | | 23.2 (20.8–26.77) | 23.3 (20.8–26.9) | 23.2 (20.7–25.4) | 0.65 |
| Nadir CD4+ (cells/mcL) | | 36 (8–83) | 43 (5–91) | 33 (14.5–77.5) | 0.98 |
| CD4+(cells/mcL) | | 406 (179–627) | 193 (111–289) | 618 (488–866) | <0.01 |
| CD4+(cells/mcL) | | | | | <0.01 |
| | <200 | 13 (28%) | 11 (50%) | 2 (8%) | |
| | ≥200–<350 | 9 (20%) | 9 (41%) | 0 (0%) | |
| | ≥350–<500 | 7 (15%) | 2 (9%) | 5 (21%) | |
| | ≥500 | 17 (37%) | 0 (0%) | 17 (71%) | |
| CD4/CD8 ratio | | 0.45 (0.22–0.81) | 0.25 (0.13–0.36) | 0.81 (0.47–0.96) | <0.001 |
| NRTI-including regimens | | | | | 0.08 |
| | No | 19 (41%) | 6 (27%) | 13 (54%) | |
| | Yes | 27 (59%) | 16 (73%) | 11 (46%) | |
| NNRTI-including regimens | | | | | 1.00 |
| | No | 31 (67%) | 15 (68%) | 16 (67%) | |
| | Yes | 15 (33%) | 7 (32%) | 8 (33%) | |
| PI-including regimens | | | | | 0.14 |
| | No | 9 (20%) | 2 (9%) | 7 (29%) | |
| | Yes | 37 (80%) | 20 (91%) | 17 (71%) | |
| INSTI-including regimens | | | | | 0.13 |
| | No | 8 (17%) | 6 (27%) | 2 (8%) | |
| | Yes | 38 (83%) | 16 (73%) | 22 (92%) | |
| Maraviroc or fostemsavir-including regimens | | | | | 0.21 |
| | No | 34 (74%) | 19 (84%) | 15 (63%) | |
| | Yes | 12 (26%) | 3 (16%) | 9 (37%) | |
| Type of ART regimen at sample collection | | | | | 0.16 |
| | ≤3 ART drugs | 25 (54%) | 9 (41%) | 16 (67%) | |
| | 4–5 ART drugs | 20 (44%) | 12 (55%) | 8 (33%) | |
| | 6–7 ART drugs | 1 (2%) | 1 (4%) | 0 (0%) | |
| Number of previous ART regimens | | 10 (5–20) | 10 (4–27) | 9 (6–17) | 0.59 |
| Number of major NRTI mutations | | 6.5 (4–9) | 7 (5–10) | 6 (4–8) | 0.24 |
| Number of major NNRTI mutations | | 3 (2–5) | 3.5 (2–5) | 2.5 (2–3.5) | 0.25 |
| Number of major PI mutations | | 7 (4–10) | 7 (4–8) | 6.5 (4.5–10.5) | 0.84 |
| Number of major INSTI mutations | | 2.5 (2–4.5) | 4 (2–5.5) | 2 (1.5–3) | 0.01 |

[a]ART = antiretroviral therapy; NRTI = nucleoside reverse transcriptase inhibitors; NNRTI = non-nucleoside reverse transcriptase inhibitors; PI = protease inhibitors; INSTI = integrase strand transfer inhibitors. The participants' characteristics at the time of sample collection were described using median [interquartile range (IQR)] or frequency (percentage).
[b]Comparisons among groups were calculated with Kruskal-Wallis test or Wilcoxon rank-sum test for continuous variables, chi-square test, or Fisher's exact test for categorical ones, as appropriate.

for the number of archived major INSTI mutations (4 [2–5.5] vs. 2 [1.5–3]; $P < 0.01$). Additional demographic, virologic, and therapeutic characteristics of 4DR-PWH at the time of sample collection and according to HIV RNA viral load are detailed in Table 1.

Of the 46 participants with PhenoSense monoclonal antibody assay results, 35 (76%) were phenotypically sensitive to TAB, 23 (50%) to ZAB, and 19 (41%) to both bNAbs, while seven (15%) were phenotypically resistant to both bNAbs. The 90% inhibitory concentration ($IC_{90}$) values for TAB and ZAB in the 4DR PWH enrolled in the study are shown in detail in Fig. 1.

Of 22 viremic participants, 19 (86%) were phenotypically sensitive to TAB, 10 (45%) to ZAB, nine (41%) to both bNAbs, and two (9%) to neither. Of the 24 participants with virologic suppression, 16 (67%) were phenotypically sensitive to TAB, 13 (54%) to ZAB, 10 (42%) to both bNAbs, and five (21%) to neither. The proportion of participants with sensitivity to both bNAbs was similar ($P = 0.99$) in viremic participants (9/22 [41%]) compared to those with virologic suppression (10/24 [42%]).

Average pairwise distance was calculated among the individual variants of HIV-1 *env* detected in plasma HIV-1 RNA and proviral DNA per participant. The HIV *env* genetic sequence diversity was high in both plasma virus (median 0.62, range 0.26–1.5) and PBMC provirus (median 1.7, range 0.56–3.6) from 4DR-PWH, consistent with diverse HIV-1 populations arising during extensive treatment history and in agreement with prior observations (18, 19) (Fig. S1).

Viral susceptibility to bNAbs was assessed by the application of HIV-1 envelope amino acid signatures known to predict sensitivity to TAB or ZAB. The analysis of phenotypic data by the presence of HIV-1 envelope signatures showed a good correlation between genotypic sensitivity predictions and observed phenotypic sensitivity, with more complex signatures predicting phenotypic susceptibility to TAB (Fig. 2; panel A) and ZAB (Fig. 2; panel B) with increasing accuracy.

The demographic, virologic, and therapeutic characteristics of 4DR-PWH at the time of sample collection and according to susceptibility to TAB and ZAB are detailed in Table 2.

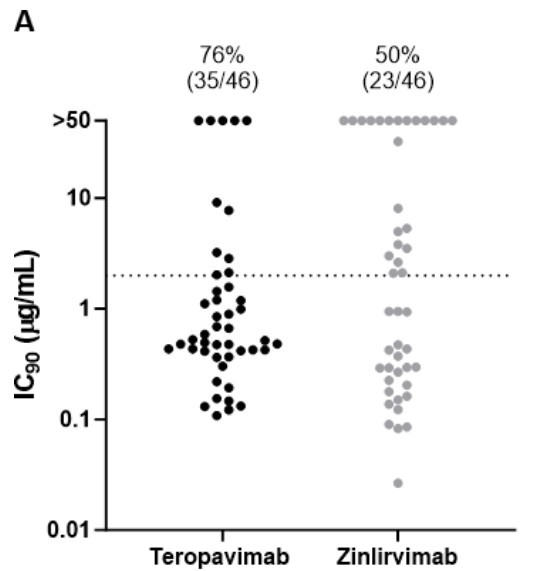
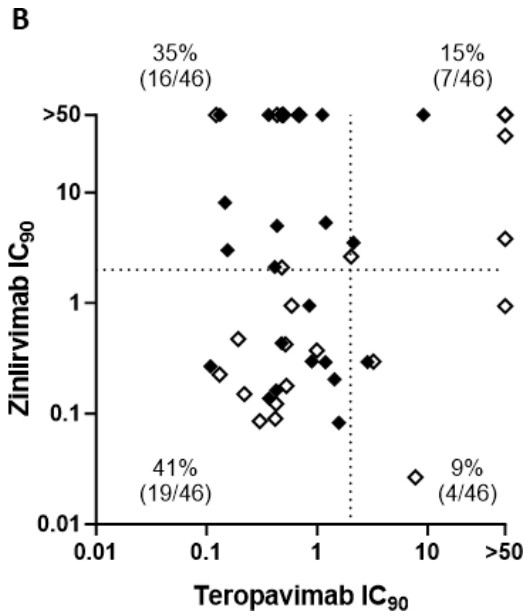

**FIG 1** Distribution of teropavimab (TAB) and zinlirvimab (ZAB) $IC_{90}$ values among PWH included in the study. Susceptibility is defined as $IC_9 \leq 2$ µg/mL (denoted by the dotted lines). (A) Percentages indicate the proportion of participants susceptible to TAB or ZAB; (B) percentages indicate the proportion of participants in each quadrant that are susceptible to TAB and ZAB (lower left quadrant), susceptible to only TAB (upper left quadrant), susceptible to only ZAB (lower right quadrant), and not susceptible to either TAB or ZAB (upper right quadrant). Solid symbols represent RNA viruses; open symbols represent DNA proviruses. bNAb, broadly neutralizing antibody; $IC_{90}$, 90% inhibitory concentration; mAb, monoclonal antibody.

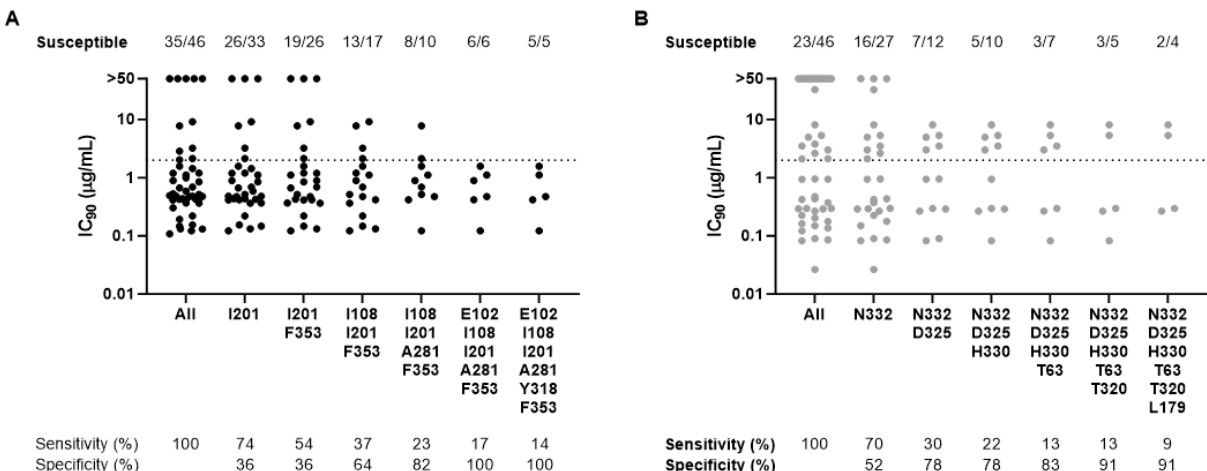

**FIG 2** Prediction of phenotypic susceptibility by genotypic signatures. RNA viruses and DNA proviruses from individual participants were plotted based on the presence of HIV-1 envelope signatures for TAB (A) and ZAB (B) sensitivity. "All" indicates all participants independent of the presence of HIV-1 envelope signatures. Dotted line represents $IC_{90}$ value = 2 µg/mL.

Non-significant correlations were observed between $IC_{90}$ values of bNAbs and age (TAB: $r = 0.08$, $P = 0.60$; ZAB: $r = -0.22$, $P = 0.15$), years of ART (TAB: $r = -0.13$, $P = 0.41$; ZAB: $r = -0.11$, $P = 0.47$), CD4+ cell count (TAB: $r = 0.17$, $P = 0.27$; ZAB: $r = -0.05$, $P = 0.75$), HIV-RNA (TAB: $r = -0.17$, $P = 0.27$; ZAB: $r = 0.09$, $P = 0.55$); marginally significant correlations were observed between TAB $IC_{90}$ values and years since HIV diagnosis at the time of sample collection ($r = 0.29$, $P = 0.05$), as well as between ZAB $IC_{90}$ values and CD8+ cell count ($r = -0.32$, $P = 0.05$).

## DISCUSSION

In our study evaluating susceptibility to TAB and ZAB in a cohort of 4DR-PWH, we observed that phenotypic susceptibility to both bNAbs was present in approximately 40% of individuals. This rate of susceptibility is similar to that observed by Selzer et al. who reported 50% susceptibility to both TAB and ZAB in individuals screened for the phase 1b proof-of-concept study that evaluated the efficacy of a combination of LEN plus TAB and ZAB in PWH with chronic infection and suppressed viremia on ART (10, 15). However, in that study, the total duration of HIV infection and the ART duration were 8.2 and 2.6 years, respectively, which is markedly lower compared to the 4DR PWH in our study (25.7 and 22.7 years, respectively).

These findings indicate that a long history of infection and ART exposure may not significantly affect viral susceptibility to bNAbs, even when associated with high viral diversity. Recent data from individuals with PHI, when viral diversity is expected to be low, revealed that the proportion of individuals susceptible to both bNAbs was 31% based on the presence of specific amino acid signatures linked to TAB and ZAB susceptibility (20); however, some individuals with virus susceptible to bNAbs may be missed due to the low sensitivity performance of these signature predictions. In other studies, higher bNAb susceptibility was associated with lower HIV-1 *env* diversity in individuals who initiated ART during acute and early HIV-1 infection vs. chronic infection, which did not differ pre- and post-treatment (19).

Furthermore, our study did not identify a correlation between co-receptor usage (CCR5 or CXCR4) at the time of sample collection and susceptibility to TAB and ZAB in accordance with the findings of other research studies (20). However, in our study, we examined a single time point, and we were, therefore, not able to evaluate the evolution of bNAbs susceptibility or HIV-1 genetic diversity over time. In addition, only nearly 25% of the PWH evaluated were receiving treatment with maraviroc or fostemsavir. While we did not observe any association between the use of these agents and bNAbs

**TABLE 2** Demographic, virologic, and therapeutic characteristics of PWH at the time of sample collection according to phenotypic sensitivity to teropavimab (TAB) and zinlirvimab (ZAB)[a]

| Variable | Category | Overall (n = 46) | With phenotypic sensitivity to TAB and ZAB (n = 19) | Without phenotypic sensitivity to TAB or ZAB (n = 27) | P-value[b] |
|---|---|---|---|---|---|
| Age (years) | | 54 (48–58) | 54 (50–57) | 55 (32–60) | 0.70 |
| Sex assigned at birth | | | | | 0.27 |
| | Female | 9 (20%) | 2 (11%) | 7 (26%) | |
| | Male | 37 (80%) | 17 (89%) | 20 (74%) | |
| HIV-1 tropism | | | | | 0.97 |
| | CCR5 | 16 (37%) | 7 (37%) | 9 (38%) | |
| | CXCR4 | 27 (63%) | 12 (63%) | 15 (62%) | |
| HIV-1 subtype | | | | | 0.69 |
| | B | 31 (91%) | 12 (92%) | 19 (90%) | |
| | CRF02_AG | 1 (3%) | 0 (0%) | 1 (5%) | |
| | F | 2 (6%) | 1 (8%) | 1 (5%) | |
| Years since HIV infection | | 25.8 (22.9–31.7) | 25.7 (22.9–29.3) | 27.5 (22.0–32.0) | 0.53 |
| Years of ART | | 23.2 (20.8–26.7) | 22.7 (20.8–24.9) | 24.9 (19.5–28.2) | 0.33 |
| Nadir CD4+ (cells/mcL) | | 36 (8–83) | 38 (9–58) | 35 (6–97) | 0.95 |
| HIV-RNA (copies/mL) | | | | | 0.96 |
| | <50 | 24 (52%) | 10 (53%) | 14 (52%) | |
| | ≥1000 | 22 (48%) | 9 (47%) | 13 (48%) | |
| CD4+ (cells/mcL) | | 406 (179–627) | 454 (173–660) | 390 (179–597) | 0.81 |
| CD4+ (cells/mcL) | | | | | 0.99 |
| | <200 | 13 (28%) | 5 (26%) | 8 (30%) | |
| | ≥200–<350 | 9 (20%) | 4 (21%) | 5 (18%) | |
| | ≥350–<500 | 7 (15%) | 3 (16%) | 4 (15%) | |
| | ≥500 | 17 (37%) | 7 (37%) | 10 (37%) | |
| CD4+/CD8+ ratio | | 0.45 (0.22–0.81) | 0.43 (0.24–0.63) | 0.47 (0.22–0.81) | 0.66 |
| NRTI-including regimens | | | | | 0.76 |
| | No | 19 (41%) | 7 (37%) | 12 (44%) | |
| | Yes | 27 (59%) | 12 (63%) | 15 (56%) | |
| NNRTI-including regimens | | | | | 0.75 |
| | No | 31 (67%) | 12 (63%) | 19 (70%) | |
| | Yes | 15 (33%) | 7 (37%) | 8 (30%) | |
| PI-including regimens | | | | | 1.00 |
| | No | 9 (20%) | 4 (21%) | 5 (19%) | |
| | Yes | 37 (80%) | 15 (79%) | 22 (81%) | |
| INSTI-including regimens | | | | | 1.00 |
| | No | 8 (17%) | 3 (15%) | 5 (19%) | |
| | Yes | 38 (83%) | 16 (85%) | 22 (81%) | |
| Maraviroc or fostemsavir-including regimens | | | | | 0.74 |
| | No | 34 (74%) | 13 (68%) | 21 (78%) | |
| | Yes | 12 (26%) | 6 (32%) | 6 (22%) | |
| Type of ART regimen at sample collection | | | | | 0.48 |
| | ≤3 ART drugs | 25 (54%) | 10 (53%) | 15 (56%) | |
| | 4–5 ART drugs | 20 (44%) | 8 (42%) | 12 (44%) | |
| | 6–7 ART drugs | 1 (2%) | 1 (5%) | 0 (0%) | |
| Number of previous ART regimens | | 10 (5–20) | 9 (7–20) | 10 (4–23) | 0.94 |
| Number of major NRTI mutations | | 6.5 (4–9) | 7 (5–10) | 6 (3–9) | 0.10 |
| Number of major NNRTI mutations | | 3 (2–5) | 2 (2–4) | 3 (2–5) | 0.26 |
| Number of major PI mutations | | 7 (4–10) | 6 (5–8) | 7 (3–11) | 1.00 |
| Number of major INSTI mutations | | 2.5 (2–4.5) | 3 (2–4) | 2 (2–5) | 0.69 |

[a]ART = antiretroviral therapy; NRTI = nucleoside reverse transcriptase inhibitors; NNRTI = non-nucleoside reverse transcriptase inhibitors; PI = protease inhibitors; INSTI = integrase strand transfer inhibitors.The participants' characteristics at the time of sample collection were described using median [interquartile range (IQR)] or frequency (percentage).

[b]Comparisons among groups were calculated with Kruskal-Wallis test or Wilcoxon rank-sum test for continuous variables, chi-square test or Fisher's exact test for categorical ones, as appropriate.

susceptibility, we cannot exclude antiviral-driven *env* evolution with an impact on TAB and ZAB susceptibility.

Our research suggests a marginal correlation between a longer history of HIV infection and higher $IC_{90}$ values for TAB. While the proportion of TAB susceptibility was similar to that observed in other studies of PWH without multidrug resistance, a longer duration of HIV infection—and consequently, exposure to ART and detectable viremia—may slightly impact TAB susceptibility. However, the PRESTIGO Registry only collects complete immunovirological data from the time the four-drug class resistance is detected. This prevents us from evaluating the possible impact of cumulative exposure to periods of detectable viremia on bNAb susceptibility. The median average pairwise distances of plasma viral and PBMC proviral sequences from 4DR PWH were 0.62 and 1.7, respectively, which are similar to or higher than those reported in previous studies that evaluated HIV-1 *env* diversity during chronic infection using a similar methodology (18, 19). It is notable that the differences in genetic diversity between plasma and PBMC virus did not correlate with the differences in phenotypic susceptibility to bNAbs. This may reflect limitations in short-read sequencing to differentiate between intact *env* necessary for successful phenotyping and defective *env* sequences that may be archived in the viral reservoir.

Susceptibility testing may be important to identify individuals more likely to respond to bNAbs. However, there is no standardized method or interpretive framework for determining susceptibility to TAB and ZAB. Here, we compared phenotypic susceptibilities in the PhenoSense mAb assay to genotypic susceptibility predictions derived from multi-position HIV-1 envelope amino acid signatures. Our study showed that more complex HIV-1 envelope amino acid signatures predicted phenotypic susceptibility with increasing specificity, consistent with prior reports (10, 21). However, a non-negligible proportion of 4DR PWH without HIV-1 envelope amino acid signatures remains susceptible to bNAbs in the phenotypic assay, indicating suboptimal sensitivity of the genotypic signature method.

Potential limitations of our study include the modest sample size and the cross-sectional design, which did not allow assessment of the evolution of TAB and ZAB susceptibility over time; however, for the first time, we have described susceptibility to bNAbs in a cohort of PWH with multidrug-resistant virus and extensive history of ART exposure and found that a relatively high percentage of these individuals retains susceptibility to these two novel long-acting agents. The confirmation of these findings in a larger sample may clarify whether heavily treatment-experienced PWH with multidrug-resistant HIV could be considered candidates for future trials evaluating bNAbs-containing regimens to achieve or maintain virologic suppression.

## MATERIALS AND METHODS

This was a multicenter, observational, cross-sectional study that used plasma or peripheral blood mononuclear cells (PBMCs) collected from 50 4DR-PWH (25 PWH with HIV-1 RNA > 1,000 copies/mL matched by age, sex assigned at birth, nadir CD4+, and years on ART to 25 virologically suppressed PWH [defined as HIV-1 RNA < 50 copies/mL]) enrolled in the PRESTIGIO Registry (NCT04098315; https://registroprestigio.org). The PRESTIGIO registry is an ongoing, observational, prospective, Italian, multicenter, annual collection of biological samples, and data on clinical, laboratory, treatment, and virological characteristics of 4DR-PWH (defined as genotypically resistant to nucleoside reverse transcriptase inhibitors [NRTIs], non-NRTIs [NNRTIs], protease inhibitors [PIs], and integrase strand transfer inhibitors [INSTIs]). Plasma and PBMC samples are collected on an annual basis from the date of enrollment and cryopreserved in a biobank (BioRep, https://www.biorep.it/). Clinical, laboratory, treatment, and virological data are collected annually from the date of evidence of 4DR (defined as baseline) (22). The PRESTIGIO Registry has been approved by the ethics committees of all participating centers, and all participants (*n* = 270 as of March 2025) have given written informed consent for their data and samples to be used for research purposes.

Phenotypic sensitivity to TAB and ZAB was determined using the PhenoSense monoclonal antibody assay from Monogram Biosciences (South San Francisco, CA, USA) with susceptibility defined as $IC_{90} \leq 2$ µg/mL for both bNAbs, consistent with previous clinical investigations (10, 15–17). Briefly, expression vectors containing plasma- or peripheral blood mononuclear cell-derived HIV-1 *env* are co-transfected with an HIV-1 genomic luciferase reporter in HEK293 cells to produce pseudovirions. Neutralizing antibody susceptibility is assessed as inhibition of pseudovirus infection of target cells following pre-incubation with bNAbs measured by luciferase activity (23).

The HIV-1 *env* gene from PhenoSense Env expression vectors was genotyped at Monogram Biosciences using Mi-Seq (Illumina, San Diego, CA, USA) next-generation sequencing. HIV-1 *env* sequences were analyzed using a previously described analysis pipeline (19). A custom-developed APOBEC hypermutation algorithm was applied to the deep sequencing data. Briefly, each read was evaluated by comparing the G->A mutations and other mutations. Reads were classified as hypermutated if they contained ≥4 G->A mutations and ≤2 non-G->A mutations. Hypermutated reads were excluded from downstream mutation analysis.

Genetic diversity within these sequences was assessed by average pairwise distance analysis using a sliding windows approach across the HIV *env* gene (19). Briefly, HIV-1 *env* was divided into 50 base pair genomic intervals with a 25-base pair overlap between two adjacent windows to reduce the impact of signal noise. The average nucleotide difference between different reads in a sliding window was calculated using Nei and Li's method (24).

Sequences were analyzed for the presence of multi-position HIV-1 Env amino acid signatures associated with *in vitro* phenotypic susceptibility to TAB and ZAB (21). Briefly, *in vitro* neutralization data combined with virus sequence information for >200 subtype B viruses were used to identify HIV Env amino acid positions important for susceptibility ($IC_{50} < 1$ µg/mL). Here, genotypic signatures were applied using the more stringent threshold of $IC_{90} \leq 2$ µg/mL. Only base-pair positions with variability <2% in viral quasi-species were considered to be part of the signature. Sensitivity of the signatures was defined as the probability that the amino acid signature was present when the virus is susceptible to the bNAb. Specificity of the signature was defined as the probability that the amino acid was not present when the virus is not susceptible to the bNAb.

## Statistical analyses

The participants' characteristics at the time of sample collection were described using median (interquartile range [IQR]) or frequency (percentage), either overall or in each group (viremic and non-viremic 4DR-PWH). Comparisons among groups were calculated with the Kruskal-Wallis test or Wilcoxon rank-sum test for continuous variables, χ test, or Fisher's exact test for categorical ones, as appropriate. The Spearman's rank test was used to test linear associations between phenotypic susceptibility and continuous clinical variables. Two-sided *P* values < 0.05 were considered statistically significant. All analyses were performed using SAS release 9.4 (SAS Institute, Cary, NC, USA).

## ACKNOWLEDGMENTS

PRESTIGIO Study Group Steering Committee: Antonella Castagna (Coordinator), Vincenzo Spagnuolo (Operations Coordinator), Daniele Armenia, Stefano Bonora, Leonardo Calza, Anna Maria Cattelan, Giovanni Cenderello, Adriana Cervo, Laura Comi, Antonio Di Biagio, Emanuele Focà, Roberta Gagliardini, Andrea Giacomelli, Filippo Lagi, Giulia Marchetti, Stefano Rusconi, Francesco Saladini, Maria Santoro, and Maurizio Zazzi. Virology Team and Biological Bank: Andrea Galli, Daniele Armenia, Francesco Saladini, Maria Santoro, Maurizio Zazzi, and BioRep SRL. Study coordinators: Elisabetta Carini, Sabrina Bagaglio, and Girolamo Piromalli. Statistical and Monitoring Team: Riccardo Lolatto and Nicolò Capra. Enrolling centers: ANCONA: Marcello Tavio, Alessandra Mataloni Paggi; AOSTA: Silvia Magnani, Manuela Colafigli AVIANO: Ornella Schioppa, Stefania Zanussi, Valentina Da Ros, Silvia Rossetto; BARI: Annalisa Saracino, Flavia Balena; BERGAMO: Laura

Comi, Daniela Valenti; BOLOGNA: Pierluigi Viale, Leonardo Calza, Federica Malerba, Silvia Cretella, Riccardo Riccardi; BRESCIA: Francesco Castelli, Emanuele Focà, Davide Minisci; BUSTO ARSIZIO: Barbara Menzaghi, Maddalena Farinazzo, Chiara Abeli; CATANIA: Bruno Cacopardo, Maurizio Celesia, Michele Salvatore Paternò Raddusa, Carmen Giarratana; CATANZARO: Paolo Fusco, Vincenzo Olivadese, Simona Mongiardi; CREMONA: Angelo Pan, Chiara Fornabaio, Paola Brambilla; FIRENZE: Alessandro Bartoloni, Filippo Lagi, Paola Corsi, Trevisan Sasha, Gasparro Giuseppe, Cecilia Costa, Alessio Bellucci, Elisa Mariabelli; FOGGIA: Teresa Santantonio, Sergio Lo Caputo, Sergio Ferrara, Arianna Narducci; GENOVA: Emanuele Pontali, Marcello Feasi, Antonio Sarà, Matteo Bassetti, Antonio Di Biagio, Sabrina Blanchi; LECCO: Stefania Piconi, Martina Bottanelli, Silvia Pontiggia, Valsecchi Giada; LEGNANO: Stefano Rusconi, Cinzia Roberta Bassoli, Francesco Bassani, Liana Bevilacqua; MILANO: Antonella Castagna, Vincenzo Spagnuolo, Camilla Muccini, Elisabetta Carini, Sabrina Bagaglio, Riccardo Lolatto, Nicolò Capra, Andrea Galli, Rebecka Papaioannu, Tommaso Clemente, Golnaz Torkjazi, Girolamo Piromalli, Spinello Antinori, Andrea Giacomelli, Tiziana Formenti, Giulia Marchetti, Lidia Gazzola, Fabiana Trionfo Fineo, Massimo Puoti, Cristina Moioli, Federico D'Amico, Simoncini Elena, Sassi Serena; MODENA: Cristina Mussini, Adriana Cervo, Giulia Nardini; NAPOLI: Elio Manzillo, Antonella Gallicchio; PADOVA: Anna Maria Cattelan, Maria Mazzitelli; PALERMO: Antonio Cascio, Marcello Trizzino; PARMA: Elisa Fronti, Diletta Laccabue, Federica Carli; PAVIA: Roberto Gulminetti, Layla Pagnucco, Mattia Demitri, Alessandra Ferrari; PERUGIA: Daniela Francisci, Giuseppe De Socio, Elisabetta Schiaroli; REGGIO EMILIA: Elisa Garlassi, Romina Corsini; ROMA: Roberta Gagliardini, Marisa Fusto, Loredana Sarmati, Vincenzo Malagnino, Tiziana Mulas, Mirko Compagno, Carlo Torti, Simona Di Giambenedetto, Silvia Lamonica, Pierluigi Salvo; SANREMO: Giovanni Cenderello, Rachele Pincino, Davide Laurenda; SASSARI: Giordano Madeddu, Andrea De Vito; SIENA: Mario Tumbarello, Massimiliano Fabbiani, Francesca Panza, Ilaria Rancan; TORINO: Giovanni Di Perri, Stefano Bonora, Micol Ferrara, Andrea Calcagno, Silvia Fantino, Giancarlo Orofino, Guido Calleri, Guastavigna Marta; and VERONA: Stefano Nardi, Marta Fiscon. Supported by: ViiV Healthcare, Gilead Sciences, Theratechnologies, and MSD.

This study was supported by a grant (CO-IT-672-6742_PRESTIGIO) from GILEAD Sciences.

A.C., L.A.V., and V.S. conceived the study, contributed to data collection, interpreted the findings, and drafted the manuscript. J.L., K.D., and C.C. contributed to the study design and data collection and reviewed the manuscript. L.G. performed the statistical analysis, contributed to data collection, and reviewed the manuscript. F.L., R.G., L.S., A.M.C., A.G., M.M.S., and M.Z. contributed to data collection and reviewed the manuscript.

## AUTHOR AFFILIATIONS

[1]Infectious Diseases, IRCCS San Raffaele Hospital, Milan, Italy

[2]School of Medicine, Vita-Salute San Raffaele University, Milan, Italy

[3]Gilead Sciences, Inc., Foster City, California, USA

[4]Infectious and Tropical Diseases Unit, Careggi University Hospital, Florence, Italy

[5]National Institute for Infectious Diseases "L. Spallanzani" IRCCS, Rome, Italy

[6]Infectious Diseases, University of Rome "Tor Vergata", Rome, Italy

[7]Infectious Diseases Unit, Department of Molecular Medicine, Padua University Hospital, Padua, Italy

[8]III Division of Infectious Diseases, ASST Fatebenefratelli Sacco, Milan, Italy

[9]Department of Experimental Medicine, University of Rome "Tor Vergata", Rome, Italy

[10]Department of Medical Biotechnology, University of Siena, Siena, Italy

## AUTHOR ORCIDs

Vincenzo Spagnuolo  http://orcid.org/0000-0002-9656-7217

## FUNDING

| Funder | Grant(s) | Author(s) |
| --- | --- | --- |
| Gilead Sciences | CO-IT-672-6742_PRESTIGIO | Antonella Castagna |

## AUTHOR CONTRIBUTIONS

Vincenzo Spagnuolo, Conceptualization, Data curation, Investigation, Validation, Visualization, Writing – original draft, Writing – review and editing | Laura Galli, Data curation, Formal analysis, Methodology, Validation, Writing – review and editing | Jiani Li, Data curation, Formal analysis, Validation, Visualization, Writing – review and editing | Keith Dunn, Data curation, Formal analysis, Validation, Writing – review and editing | Filippo Lagi, Validation, Visualization, Writing – original draft | Roberta Gagliardini, Validation, Visualization, Writing – review and editing | Loredana Sarmati, Validation, Visualization, Writing – review and editing | Anna Maria Cattelan, Validation, Visualization, Writing – review and editing | Andrea Giacomelli, Validation, Visualization, Writing – review and editing | Maria Mercedes Santoro, Data curation, Validation, Visualization, Writing – review and editing | Maurizio Zazzi, Data curation, Validation, Visualization, Writing – review and editing | Christian Callebaut, Data curation, Investigation, Validation, Visualization, Writing – review and editing | Antonella Castagna, Conceptualization, Data curation, Investigation, Supervision, Validation, Visualization, Writing – review and editing | Laurie A. VanderVeen, Conceptualization, Data curation, Formal analysis, Investigation, Validation, Visualization, Writing – review and editing

## DATA AVAILABILITY

The viral sequences investigated in our study have been released on the GenBank database with the following accession numbers: PX254985-PX255030.

## ADDITIONAL FILES

The following material is available online.

### Supplemental Material

**Fig. S1 (Spectrum02777-24-s0001.tif).** HIV Env sequence diversity in PWH.
**Supplemental material (Spectrum02777-24-s0002.docx).** Fig. S1 legend.

### Open Peer Review

**PEER REVIEW HISTORY (review-history.pdf).** An accounting of the reviewer comments and feedback.

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
