## [Reviewer comments · Microbiology Spectrum]

Microbiology Spectrum

TEROPAVIMAB AND ZINLIRVIMAB SENSITIVITY IN PEOPLE LIVING WITH MULTIDRUG RESISTANT HIV-1: DATA FROM THE PRESTIGIO REGISTRY

Vincenzo Spagnuolo, Laura Galli, Jiani Li, Keith Dunn, Filippo Lagi, Roberta Gagliardini, Loredana Sarmati, Anna Cattelan, Andrea Giacomelli, Maria Mercedes Santoro, Maurizio Zazzi, Christian Callebaut, Antonella Castagna, and Laurie VanderVeen

Corresponding Author(s): Vincenzo Spagnuolo, IRCCS Ospedale San Raffaele

Review Timeline:

Submission Date:	January 13, 2025
Editorial Decision:	May 19, 2025
Revision Received:	July 1, 2025
Accepted:	July 29, 2025

Editor: Jose Martinez-Navio

Reviewer(s): The reviewers have opted to remain anonymous.

Transaction Report:

DOI: <https://doi.org/10.1128/spectrum.02777-24>

Re: Spectrum02777-24 (TEROPAVIMAB AND ZINLIRVIMAB SENSITIVITY IN PEOPLE LIVING WITH MULTIDRUG RESISTANT HIV-1: DATA FROM THE PRESTIGIO REGISTRY)

Dear Dr. Vincenzo Spagnuolo:

Thank you for the privilege of reviewing your work. Below you will find my comments, instructions from the Spectrum editorial office, and the reviewer comments.

Revision Guidelines

Sincerely,
Jose Martinez-Navio
Editor
Microbiology Spectrum

Reviewer #1 (Comments for the Author):

This is a focused investigation and concisely written manuscript to describe phenotypic sensitivity to two bNABs with particular clinical relevance (ZAB and TAB). I felt the study design also usefully concentrated on individuals who were heavily treatment experienced, as well as providing data for both viremic and aviremic individuals

(plus working to have participant characteristics matched between the groups).
In addition, I felt the Figures were nicely informative.

However, there are a number of questions and comments I had, primarily to make the presentation more clear and informative. Please see detailed comments below.

[1] Abstract line 59-60, and Results line 171.

Years since HIV diagnosis has a positive correlation with phenotypic sensitivity to TAB (noted as correlation with 'phenotypic susceptibility' in Statistical Analysis section) And Discussion "longer history of HIV infection and an increased susceptibility to bNAbs." I am not clear about this positive correlation. As Intro highlights anticipated negative association with susceptibility "In addition, selective immune pressure during untreated infection may drive HIV-1 envelope evolution, resulting in viral populations that can evade recognition by bNAbs and limit their effectiveness."

Are these correlations with IC90 values (Spearman) ?

-where high IC90 values indicate *low* sensitivity/susceptibility.

Or with 'susceptibility' (defined as IC90 < 2 mcg/mL -perhaps analyzing as yes(1) vs. no(0))

The basic directionality of these presented associations is not clear (final paragraph Results).

If the Spearman correlations are with actual IC90 values, then please rewrite Results with this language. (and re-assess/re-examine corresponding Discussion to be optimally clear to the reader)

[2] Related: Is this variable the years since HIV diagnosis at time of sampling ?

A more interpretable variable may be 'years of viremia'

(pre-ART years + years of viremia after ART initiation)

However, Discussion notes the authors having incomplete history of viral suppression so appears this variable and analysis would not be possible.

The number of years since HIV diagnosis at time of ART initiation might also be considered. (years of untreated HIV)

Please clarify why this variable ('years since HIV diagnosis' at sample collection) was chosen and especially regarding its interpretation, as it would seem to include and count years while virally suppressed on ART.

[3] Related:

Discussion, line 200.

Here the association with greater # years of HIV infection is interpreted "among those with controlled viremia"

However, final paragraph of Results does not have this indication and so reads as being the correlation in the full study population (both viremia and aviremic groups combined).

Please assess and clarify (including above questions re directionality).

[4] Methods, line 250.

Please could the authors provide background/reference for the use of <2 mcg/mL threshold to define susceptibility.

(for example, it appears this same threshold was used as eligibility criterion in Eron et al. 2024 Ref #15, so appears could usefully cite this TAB+ZAB clinical study here, as well as the related Selzer et al. Ref #10]

[5] Recommend edit throughout to be more careful about significant digits in the presentation of the results

eg Recommend put correlations to 2 digits: $r = 0.29$ and $r = -0.32$ (Abstract and Results)

And p-values ≥ 0.10 as 2 digits (eg $p=0.60$)

And eg Results 1st paragraph:

Age (median, quartiles) as integers

And in particular, the N is well below 100. The study cannot estimate percentages with precision to tenth of percentage.

80.4% male => 80% male. 91% subtype B.

and all %s in Table 1 to not show decimal percentages.

Additional.

*Line 134 and Table 1 "previous ART lines" Do you mean "previous ART regimens" ?

As well: would put median,Q1,Q3 as integers in Table 1. Isn't this data item inherently integer valued ?

*Figure 1 legend. recommend replace (2x) 'ratio' with 'proportion' => "indicate proportion of participants"

*Figure 1B legend. It appears that text should be:

"susceptible to only TAB (upper left quadrant)" and

"susceptible to only ZAB (lower right quadrant)"

Please check/update (appears 'ZAB' vs 'TAB' needs updating)

Reviewer #2 (Comments for the Author):

This work investigated the sensitivity to two broadly neutralizing antibodies (teropavimab (TAB) and zinlirvimab (ZAB) in an Italian cohort harboring viruses heavily resistant to HIV antiretroviral therapy (4 class resistance.)

They used the PhenoSense monoclonal antibody assay (Monogram). They performed NGS on plasma (when virus was detectable) and on proviral DNA.

They did not find differences in sensitivity to both bNABS between viremic and non viremic PWH.

Of the 46 participants tested with the PhenoSense assay 76% were phenotypically sensitive to TAB, 50% to ZAB, and 41% to both bNABs. 15% were phenotypically resistant to both bNABs.

Of the 22 viremic PWH, 86% were phenotypically sensitive to TAB, 45% to ZAB, 41% to both and 9% to neither. Of the 24 PWH and virologic suppression, 67% were phenotypically sensitive to TAB, 54% to ZAB, 42% to both and 21% to neither.

Double sensitivity did not differ between viremic and non-viremic people. Diversity was not statistically different between viremic and noneviremic patients, however, it was larger from proviral DNA generated sequences, probably reflecting the archived diversity over time with all the defective viruses. This could be stated somehow a bit in more detail.

This is an interesting paper of importance to the field. It is well done, representing a rare population (well documented 4 class resistance cases are relatively rare). The methods used are state of the art, analysis is well done.

Just one question: were proviral DNA sequences corrected for APOBEC 3G/F hypermutations, which potentially may also affect sensitivity in env. If not, a parallel analysis should be performed taking into account APOBEC-mutations. (of course there is no clear-cut threshold but still, if not corrected resistance against those antibodies might potentially be overestimated.) If it was done then it should also be mentioned and by what method the env sequences were filtered for APOBEC mutations. Recently, it has been recognized that APOBEC mutations should be considered when interpreting proviral DNA testing in general (Wensing et al, CID, PMID: 40176204)

One thing is clear as has been shown in other papers:

Before using TAB and ZAB resistance testing before using those bNABs is necessary. Maybe they could state this a bit more clearly.

REVIEWERS' COMMENTS (Manuscript reference number: Spectrum02777-24)

Title: Teropavimab and Zinlirvimab Sensitivity in People Living with Multidrug Resistant HIV-: Data from the PRESTIGIO Registry

Reviewer #1:	
Reviewer Comments:	Response:
This is a focused investigation and concisely written manuscript to describe phenotypic sensitivity to two bNAbs with particular clinical relevance (ZAB and TAB). I felt the study design also usefully concentrated on individuals who were heavily treatment experienced, as well as providing data for both viremic and aviremic individuals (plus working to have participant characteristics matched between the groups). In addition, I felt the Figures were nicely informative. However, there are a number of questions and comments I had, primarily to make the presentation more clear and informative. Please see detailed comments below.	The authors appreciate the reviewer's recognition on the quality the work presented and importance of this data, as well as the constructive review of the manuscript. Detailed point-by-point responses are provided below.
Specific Comments:	Response:
[1] Abstract line 59-60, and Results line 171. Years since HIV diagnosis has a positive correlation with phenotypic sensitivity to TAB (noted as correlation with 'phenotypic susceptibility' in Statistical Analysis section) And Discussion "longer history of HIV infection and an increased susceptibility to bNAbs." I am not clear about this positive correlation. As Intro highlights anticipated negative association with susceptibility "In addition, selective immune pressure during untreated infection may drive HIV-1 envelope evolution, resulting in viral populations that can evade recognition by bNAbs and limit their effectiveness."	We would like to thank the reviewer for this useful comment. There is an association between the duration of HIV infection and IC90 values. This association is positive; therefore, the longer the duration of HIV infection, the higher the IC90 values and the lower the susceptibility to TAB. As suggested, we clarified this in the Results section and appropriately revised the Discussion section.

Are these correlations with IC90 values (Spearman) ? -where high IC90 values indicate *low* sensitivity/susceptibility. Or with 'susceptibility' (defined as IC90 < 2 mcg/mL -perhaps analyzing as yes(1) vs. no(0)) The basic directionality of these presented associations is not clear (final paragraph Results). If the Spearman correlations are with actual IC90 values, then please rewrite Results with this language. (and re-assess/re-examine corresponding Discussion to be optimally clear to the reader)	
Related: Is this variable the years since HIV diagnosis at time of sampling? A more interpretable variable may be 'years of viremia' (pre-ART years + years of viremia after ART initiation) However, Discussion notes the authors having incomplete history of viral suppression so appears this variable and analysis would not be possible. The number of years since HIV diagnosis at time of ART initiation might also be considered. (years of untreated HIV) Please clarify why this variable ('years since HIV diagnosis' at sample collection) was chosen and especially regarding its interpretation, as it would seem to include and count years while virally suppressed on ART.	The authors were referring to the years since HIV diagnosis at the time of sample collection. Text has been modified for clarity (Line 170 in marked-up manuscript). Additionally, the authors agree that it would be interesting to understand the impact of time of viremia on bNAb susceptibility. However, as these participants are heavily treatment experienced, the number of regimens/failures they have been on are difficult to accurately quantify with historical records, as well as the duration of untreated HIV, or viremia during living with HIV. In the PRESTIGIO Registry, complete immunovirological data are requested only from the date of four-drug class resistance. Consequently, while this would be of interest, the association between cumulative detectable viremia exposure and bNAbs susceptibility cannot be assessed with the current dataset.
Discussion, line 200. Here the association with greater # years of HIV infection is interpreted "among those with controlled viremia" However, final paragraph of Results does not have this indication and so reads as being the correlation in the full study population (both viremia and aviremic groups combined). Please assess and clarify (including above	In light of this and the first comment, we have modified the discussion section.

questions re directionality).	
Methods, line 250. Please could the authors provide background/reference for the use of <2 mcg/mL threshold to define susceptibility. (for example, it appears this same threshold was used as eligibility criterion in Eron et al. 2024 Ref #15, so appears could usefully cite this TAB+ZAB clinical study here, as well as the related Selzer et al. Ref #10]	The susceptibility criteria in the current study were designed to be consistent with eligibility criteria used in clinical studies of TAB and ZAB. We have included the following text (Line 246-248 in the marked-up manuscript) along with the below references to address this comment: Phenotypic sensitivity to TAB and ZAB was determined using the PhenoSense Monoclonal Antibody assay from Monogram Biosciences (South San Francisco, CA, USA) with susceptibility defined as $IC_{90} \leq 2$ mcg/mL for both bNAbs, consistent with previous clinical investigations  1. Eron et al. 2024 Ref #15 2. Selzer et al. Ref #10 3. Eron et al. 2025 (updated reference #17) 4. Ogbuagu, O. et al. 32nd CROI, Mar 09-12, 2025. San Francisco, CA, USA, Oral abstract 151 [reference #16]
Recommend edit throughout to be more careful about significant digits in the presentation of the results eg Recommend put correlations to 2 digits: $r = 0.29$ and $r = -0.32$ (Abstract and Results) And p-values ≥ 0.10 as 2 digits (eg $p=0.60$) And eg Results 1st paragraph: Age (median, quartiles) as integers And in particular, the N is well below 100. The study cannot estimate percentages with precision to tenth of percentage. 80.4% male => 80% male. 91% subtype B. and all %s in Table 1 to not show decimal percentages.	The authors agree with the reviewer and have updated the values throughout.
Line 134 and Table 1 "previous ART lines" Do you mean "previous ART regimens" ? As well: would put median,Q1,Q3 as integers in Table 1. Isn't this data item inherently integer valued?	As suggested, we replaced "number of previous ART lines" with "number of previous ART regimens" and reported the median and Q1, Q3 as integers.
Figure 1 legend. recommend replace (2x) 'ratio' with 'proportion' => "indicate proportion of	The Figure 1 legend has been updated to state "proportion".

participants"	
Figure 1B legend. It appears that text should be: "susceptible to only TAB (upper left quadrant)" and "susceptible to only ZAB (lower right quadrant)" Please check/update (appears 'ZAB' vs 'TAB' needs updating)	This has been corrected in the Figure 1B legend.
Reviewer #2:	
Reviewer Comments:	Response:
This work investigated the sensitivity to two broadly neutralizing antibodies (teropavimab (TAB) and zinlirvimab (ZAB) in an Italian cohort harboring viruses heavily resistant to HIV antiretroviral therapy (4 class resistance.) They used the PhenoSense monoclonal antibody assay (Monogram). They performed NGS on plasma (when virus was detectable) and on proviral DNA. They did not find differences in sensitivity to both bNABS between viremic and non viremic PWH. Of the 46 participants tested with the PhenoSense assay 76% were phenotypically sensitive to TAB, 50% to ZAB, and 41% to both bNABs. 15% were phenotypically resistant to both bNABs. Of the 22 viremic PWH, 86% were phenotypically sensitive to TAB, 45% to ZAB, 41% to both and 9% to neither. Of the 24 PWH and virologic suppression, 67% were phenotypically sensitive to TAB, 54% to ZAB, 42% to both and 21% to neither. Double sensitivity did not differ between viremic and non-viremic people. Diversity was not statistically different between viremic and noneviremic patients, however, it was larger from proviral DNA generated sequences, probably reflecting the archived diversity over	The authors appreciate the reviewer's comments of the importance of this data for the field, as well as the detailed review of the manuscript. Detailed point-by-point response are provided below.

time with all the defective viruses. This could be stated somehow a bit in more detail. This is an interesting paper of importance to the field. It is well done, representing a rare population (well documented 4 class resistance cases are relatively rare). The methods used are state of the art, analysis is well done.	
Specific Comments:	Response:
Just one question: were proviral DNA sequences corrected for APOBEC 3G/F hypermutations, which potentially may also affect sensitivity in env. If not, a parallel analysis should be performed taking into account APOBEC-mutations. (of course there is no clear-cut threshold but still, if not corrected resistance against those antibodies might potentially be overestimated.) If it was done then it should also be mentioned and by what method the env sequences were filtered for APOBEC mutations. Recently, it has been recognized that APOBEC mutations should be considered when interpreting proviral DNA testing in general (Wensing et al, CID, PMID: 40176204)	The authors agree with the reviewer that this is relevant to include and have added the following wording to the methods section (Lines 255-259 in the marked-up manuscript) to clarify the process: A custom-developed APOBEC hypermutation algorithm was applied to the deep sequencing data. Briefly, each read was evaluated by comparing the G->A mutations and other mutations. Reads were classified as hypermutated if they contained ≥ 4 G->A mutations and ≤ 2 non-G->A mutations. Hypermutated reads were excluded from downstream mutation analysis. ‘
One thing is clear as has been shown in other papers: Before using TAB and ZAB resistance testing before using those bNABs is necessary. Maybe they could state this a bit more clearly.	The following text has been included to address this comment (Lines 108-109 and 212 in the marked-up manuscript): Introduction: Thus, bNABs susceptibility testing before initiating treatment with TAB and ZAB may improve clinical success Discussion: Susceptibility testing may be important to identify individuals more likely to respond to bNABs.

Re: Spectrum02777-24R1 (TEROPAVIMAB AND ZINLIRVIMAB SENSITIVITY IN PEOPLE LIVING WITH MULTIDRUG RESISTANT HIV-1: DATA FROM THE PRESTIGIO REGISTRY)

Dear Dr. Vincenzo Spagnuolo:

Your manuscript has been accepted. Please do your best to incorporate the suggestions from Reviewer #1. I am forwarding your manuscript to the ASM production staff for publication. Your paper will first be checked to make sure all elements meet the technical requirements. ASM staff will contact you if anything needs to be revised before copyediting and production can begin. Otherwise, you will be notified when your proofs are ready to be viewed.

Sincerely,
Jose Martinez-Navio
Editor
Microbiology Spectrum

Reviewer #1 (Comments for the Author):

I appreciate the responses to reviewers and the efforts to update the manuscript.

[1] One item I see is that I would recommend that the Abstract (line 59-61) be similarly updated to match final paragraph Results.
ie to describe correlations between "IC90" (not 'phenotypic sensitivity') and covariates (years since HIV diagnosis and CD8 count, respectively).

[2] I also do not see that Table 1 or Table 2 state what the summary numbers represent. eg, are these 'Median (Q1 - Q3)' ?
For completeness, recommend add as footnote to both Tables.
And to Results line 132 (expand to note Q1-Q3 also here at first use in text, for age).
Currently just states median but three numbers are shown.

*Line 204. Extra period.

Reviewer #2:

Reviewer #2 had recommended acceptance during the previous round of reviews and the Editor has double-checked the

positive responsiveness of the authors to this Reviewer's comments.